# Oxygen Saturation Measurements from Green and Orange Illuminations of Multi-Wavelength Optoelectronic Patch Sensors

**DOI:** 10.3390/s19010118

**Published:** 2018-12-31

**Authors:** Samah Alharbi, Sijung Hu, David Mulvaney, Laura Barrett, Liangwen Yan, Panagiotis Blanos, Yasmin Elsahar, Samuel Adema

**Affiliations:** 1Wolfson School of Mechanical, Electrical and Manufacturing Engineering, Loughborough University, Ashby Road, Loughborough, Leicestershire LE11 3TU, UK; S.M.Alharbi@lboro.ac.uk (S.A.); D.J.Mulvaney@lboro.ac.uk (D.M.); P.Blanos@lboro.ac.uk (P.B.); Y.Elsahar@lboro.ac.uk (Y.E.); S.T.Adema@lboro.ac.uk (S.A.); 2School of Sport, Exercise and Health Sciences, Loughborough University, Ashby Road, Loughborough, Leicestershire LE11 3TU, UK; L.A.Barrett@lboro.ac.uk; 3School of Mechatronic Engineering and Automation, Shanghai University, Shanghai 200072, China; lw_yan@staff.shu.edu.cn

**Keywords:** oxygen saturation (SpO_2_), green and orange illuminations, optoelectronic patch sensor (mOEPS), pulse oximetry, physical movement

## Abstract

Photoplethysmography (PPG) based pulse oximetry devices normally use red and infrared illuminations to obtain oxygen saturation (SpO${}_{2}$) readings. In addition, the presence of motion artefacts severely restricts the utility of pulse oximetry physiological measurements. In the current study, a combination of green and orange illuminations from a multi-wavelength optoelectronic patch sensor (mOEPS) was investigated in order to improve robustness to subjects’ movements in the extraction of SpO${}_{2}$ measurement. The experimental protocol with 31 healthy subjects was divided into two sub-protocols, and was designed to determine SpO${}_{2}$ measurement. The datasets for the first sub-protocol were collected from 15 subjects at rest, with the subjects free to move their hands. The datasets for the second sub-protocol with 16 subjects were collected during cycling and walking exercises. The results showed good agreement with SpO${}_{2}$ measurements (*r* = 0.98) in both sub-protocols. The outcomes promise a robust and cost-effective approach of physiological monitoring with the prospect of providing health monitoring that does not restrict user physical movements.

## 1. Introduction

Many researchers have contributed to improving the performance of physiological monitoring systems, including the simplification of the setting-up of equipment, generating more accurate readings or reducing the cost of obtaining measurements. One of the monitoring approaches commonly used is photoplethysmography (PPG), a non-invasive optical technique able to measure volume changes in blood vessels and to estimate vital health signs, such as heart rate (HR), heart rate variability, respiration rate, blood pressure and oxygen saturation (SpO${}_{2}$) [1]. PPG sensors are normally designed to operate either in transmission mode or reflectance mode. In transmission mode, a photo detector (PD) is placed to detect light after it has been transmitted through a subject’s body part (such as a finger or earlobe). In a reflectance mode, the PD is usually located on the same side of the body part as the illumination source and captures back-scattered light from skin and tissue. Transmission mode is widely used in hospital and clinical settings, such as in pulse oximetry, whereas the reflectance approach is mainly used in wearable sensors, such as smart watches. To date, a number of wearable sensors used to detect physiological parameters have been developing commercially, such as HRM-3200 (H3, Daejeon, Korea) [2], Honour S1 (Huawei, Shenzhen China) [3], MZ-3Belt (Myzone, Nottingham, UK) [4], NoninWristOx 3100 [5], AMON (Advanced Medical Monitor) physiological monitor [6] LifeShirt [7], Smart 32 Shirt [8], SenseWear [9] and Smart Vest [10]. The transmission mode is restrictive in terms of the number of options available for measurement sites, e.g., finger, earlobe and toe, in addition to the limitation of the patients’ freedom of movement [11]. For example, fingertip sensors, such as those used for pulse oximetry, interfere with daily activities and it can be difficult to measure vital signs reliably during hand movements. By using a reflectance multi-wavelength optoelectronic sensor (mOEPS) in this work, the number of suitable sites available on the body increases dramatically. Furthermore, it has been reported that the reflectance mode PPG sensor probe MaxFast (NellcorTM, Minneapolis, MN, USA) has been used clinically to provide non-invasive continuous monitoring, albeit giving occasional erroneous readings [11]. An mOEPS was applied to carry out a specific study using green and orange illuminations. The raw reflected PPG signal is often considered to be composed of two separate components, namely a direct current (DC) and an alternating current (AC) [11].

The DC component arises from non-pulsatile tissue such as bones and veins that maintain a constant absorption characteristic during measurement. The AC component is the variation in the intensity of reflected light that results from variations in absorption characteristics as the blood pulses through tissue. The amplitude of the AC component is typically only 1% to 2% of that of the DC component [12], making its extraction more susceptible to the presence of measurement and electrical noise. In this study, we proposed that the most encouraging research area would consider reflectance mode operation, as this approach is more likely to provide comfort and convenience for end users.

Current PPG based pulse oximetry sensors mainly use two separate wavelength illuminations, namely red and infrared (IR). However, these two wavelength illuminations have been reported to produce signals of poor reliability during the movements [13]. It would be appropriate to consider green as an alternative illumination colour as it has been reported to provide signals with improved resilience to motion artefacts [11]. In addition, the measurement of certain physiological parameters, such as SpO${}_{2}$, requires that at least two separate signals be obtained from incident light of different wavelengths. Many less expensive health monitoring products, e.g., smartwatches, tend to use only one wavelength and are consequently restricted to HR measurement only.

In this study, an mOEPS with at least four wavelength illumination sources was used to multiply illumination on a designated tissue and to extract SpO${}_{2}$ from same device. Conventionally, pulse oximeters with red and IR illuminations are used to measure SpO${}_{2}$ non-invasively as haemoglobin is the oxygen-transport metalloprotein in red blood cells [14] and the light absorption of oxy-haemoglobin (HbO${}_{2}$) relative to that of deoxyhaemoglobin (Hb) is dependent on the illumination wavelength. Under red illumination, the absorption of Hb is greater than that of HbO${}_{2}$, whereas the relative absorption is reversed under IR light. By measuring the absorption by the blood of both red and IR light, it is possible to estimate the value of SpO${}_{2}$. As tissues in the body generally contain water, they generally absorb light more strongly in the longer wavelength IR region [12], whereas melanin is known to exhibit greater absorption of light at shorter wavelengths [11]. Red blood cells are reported to have greater light absorption at red and near IR wavelengths [15] and this could be used to facilitate the measurement of blood volume changes. Consequently, red and IR illuminations are commonly used as illuminated light sources in pulse oximetry [16]. There are several studies that have used red light (650–750 nm) [17] and IR light (850–1000 nm) [18] to measure SpO${}_{2}$ using the PPG technique [18]. However, PPG signals are easily affected by the sources of noise from human tissue deformation, ambient light interference and electromagnetic field, thus making good readings of physiological parameters difficult, as anticipated in practice [19].

In this study, 525 nm (green) and 595 nm (orange) wavelength illuminations were investigated as alternative illuminated light sources to be used in the extraction of SpO${}_{2}$ reading. Green and orange illuminations are absorbed closer to the surface of the skin in comparison with the depth of absorption of incident red and IR light. Consequently, although green and orange illumination penetrate the skin to a lesser degree, their signal-to-noise ratio is improved as they suffer from less attenuation due to their comparatively shorter path lengths [20]. Figure 1 shows the skin penetration depths of light signals at wavelengths in the range of 400 nm to 1000 nm.

Figure 2 shows that there is a sufficiently significant difference in the absorption of Hb and HbO${}_{2}$ at each of the green and orange wavelengths to make the method suitable for extracting SpO${}_{2}$ measurements.

The aim of the study is to research how to effectively work out SpO${}_{2}$ by means of mOEPS with its unique multi-wavelength illumination functionality. As the mOEPS has four wavelengths, e.g., green (525 nm), orange (595 nm), red (650 nm) and near Infrared (IR) (870 nm), the green and orange illuminations were used as primarily results to extract SpO${}_{2}$ [23]. The study was specified to be able to operate under a number of different scenarios including physical movements [24]. To achieve relevant quality signals, the mOEPS was able to be located at various measurement sites of subjects [25] who had a range of skin pigments [26].

The specific objectives of the paper can be stated as:To study how effectively to obtain physiological parameters SpO${}_{2}$ from mOEPS using green and orange illuminations.To evaluate the performance of mOEPS against a pulse oximetry through implementation of a setup protocol involving healthy subjects.To compare the performance of different illuminations (green and orange, red and infrared) from mOEPS to work out the SpO${}_{2}$ measurement.

## 2. Methodology

This section describes the approach taken in the current work in the measurement of SpO${}_{2}$ and provides a description of the sensing and processing systems.

### 2.1. Experimental Protocol

Thirty-one (31) healthy subjects (gender: male, age: 25 ± 5 years, height: 179 cm ± 4 cm, and Body Mass Index: 22 ± 1 kg/m^2^) volunteered to take part in the study. The experimental protocol (R18-P006) was approved by the Loughborough University Ethical Advisory Committee on 19 February 2018. The protocol was designed into two different groups of 15 and 16 subjects as two sub-protocols, respectively. Both sub-protocols were used to investigate the performance of mOEPS in the extraction of SpO${}_{2}$ in sitting mode and during physical activities. For the first sub-protocol, the subjects were sitting and gently moving their hands and the green and orange illumination were compared and validated against the commercial pulse oximetry for extracting SpO${}_{2}$. For the second sub-protocol, the subjects completed the cycling and walking activities and the mOEPS was solely used to compare the SpO${}_{2}$ of different wavelengths during exercises. All the subjects in both sub-protocols were required to abstain from drinking alcohol, coffee or other substances that may have affected their performance in the 24 h leading up to the taking of the measurements. The subjects’ data was recorded using the patch sensor placed in the palm of the left hand. The subjects were free to move their hands as usual. In order to collect benchmark data, TempIRTM pulse oximetry (Shenzhen Jumper Medical Equipment Co., Ltd., Shenzhen, China), which uses optical PPG techniques, was utilized to match with our mOEPS in the first sub-protocol. The commercial pulse oximeter was used to obtain a reference SpO${}_{2}$ parameter for comparison against the mOEPS obtained parameters. The data recordings from the commercial sensor and the mOEPS were processed and analysed using a bespoke MATLAB programs (R2016b) (MathWorks, Cambridge, UK). The subjects were first asked to remain in a seated, upright position for one minute while the sensors were attached, before the raw data was captured over a period of three minutes during free hand movement in the first sub-protocol. However, in the second sub-protocol, the subjects were asked to perform different activities, such as cycling (20 rpm) and walking (5 km/h), for a period of four minutes per activity. A time window of 10 s duration was used in the calculation SpO${}_{2}$. Figure 3 shows typical time duration for the captured data.

### 2.2. Extraction of Oxygen Saturation (SpO${}_{2}$)

The oxygen saturation of the blood is the degree to which oxygen is chemically combined with haemoglobin. SpO${}_{2}$ can be worked out as the fractional ratio between the concentration of oxygenated haemoglobin and the total haemoglobin present in the blood, as shown in the following Equation (Equation 1):(1)$${\mathrm{SpO}}_{2}\left(\%\right)=\frac{\left[{\mathrm{HbO}}_{2}\right]}{\left(\left[{\mathrm{HbO}}_{2}\right]+\left[\mathrm{Hb}\right]\right)}\times 100\%,$$
where [Hb] is the concentration of the deoxyhaemoglobin form and [HbO${}_{2}$] is the concentration of the oxy-haemoglobin form.

According to Lambert Beer’s law [12], the transmission of illumination decays exponentially with the absorption coefficient and the optical path length. The tissues absorb light differently during the diastolic and systolic phases. In this study, green and orange wavelengths were used to extract the SpO${}_{2}$ and their absorption is expressed in the following Equations (2) and (3):(2)$$I_{h,G}=I_{in,G}e^{-\mu \left(\lambda_{G}\right)d_{Gmin}},$$
(3)$$I_{l,G}=I_{in,G}e^{-\mu \left(\lambda_{G}\right)d_{Gmax}},$$
where $I_{h,G}$ and $I_{l,G}$ are the intensity of the green light transmission during the diastolic phase and systolic phases, respectively, $I_{in,G}$ is the intensity of incident green light, μ is the absorption coefficient, λ is the wavelength of the green light (525 nm) and $d_{Gmin}$ and $d_{Gmax}$ are the minimum and maximum optical path lengths, respectively.

By taking the logarithm of the ratio of transmissions in the diastolic and systolic phases, and, assuming the absorption coefficient is known, the difference between the optical path lengths for the green illumination can be determined by Equation (Equation 4);
(4)$$\ln\left(\frac{I_{h,G}}{I_{l,G}}\right)=-\mu \left(\lambda_{G}\right)\left(d_{Gmax}-d_{Gmin}\right)=-\mu \left(\lambda_{G}\right)\Delta d_{G}.$$

Similarly, for the orange illumination, the difference between the optical path lengths is given by Equation (Equation 5);
(5)$$\ln\left(\frac{I_{h,O}}{I_{l,O}}\right)=-\mu \left(\lambda_{O}\right)\left(d_{Omax}-d_{Omin}\right)=-\mu \left(\lambda_{O}\right)\Delta d_{O}.$$

The absorption ratio of green and orange wavelengths (*R*) can be calculated as follows of Equation (Equation 6);
(6)$$R=\frac{\ln(\frac{I_{h,G}}{I_{l,G}})}{\ln(\frac{I_{h}{,}_{O}}{I_{l}{,}_{O}})}=\frac{\mu \left(\lambda_{G}\right)\Delta d_{G}}{\mu \left(\lambda_{O}\right)\Delta d_{O}}.$$

From the previous equation, ln($I_{h},$${}_{G}$/$I_{l},$${}_{G}$) is the arterial transmittance variation under incident green light $\mu (\lambda_{G})$ and the arterial transmittance variation under incident orange light $\mu (\lambda_{O})$. If it is assumed that both sources of illumination (green and orange) are placed at the same position on the palm, then the optical path lengths (light travel to the skin) are approximately the same ($\Delta d_{G}$ ≈ $\Delta d_{O}$). The lengths of the green and orange return optical paths from the arterial blood flow to the detector will also be similar. As a result, the absorption ratio depends principally on the relative absorption coefficients obtained at the two wavelengths and is largely independent of the optical path lengths:(7)$$R=\frac{\mu (\lambda_{G})}{\mu (\lambda_{O})}.$$

In general, SpO${}_{2}$ can also be extracted from signals captured by PPG sensors with two different wavelength illuminations and one photo sensor. In the current work, the green and orange illumination sources were used for both signals of two different wavelength measurements obtained from the DC and AC components, which represent the absorption light. The DC components can be extracted from the raw data as shown in Figure 4a, and the AC amplitude can be determined from the positions of peaks and troughs as shown in Figure 4b.

The absorption ratio can be estimated from a ‘ratio of ratios’ measurement as follows:(8)$$R=\frac{\mu (\lambda_{G})}{\mu (\lambda_{O})}=\frac{\left(\frac{{\mathrm{AC}}_{green}}{{\mathrm{DC}}_{green}}\right)}{\left(\frac{{\mathrm{AC}}_{orange}}{{\mathrm{DC}}_{orange}}\right)}.$$

The oxygen saturation can then be calculated from Equation (Equation 8);
(9)$${\mathrm{SpO}}_{2}=A-BR,$$
where A and B are coefficients that are obtained from a best-fit linear regression equation relating to the standard pulse oximeter series reading against the ratio R of mOEPS. Table 1 shows the ratio values (*R*) obtained from the mOEPS and the SpO${}_{2}$ values obtained from the pulse oximeter.

### 2.3. Experimental Setup

The mOEPS was developed to capture PPG signals at green, orange, red and IR illuminations. A microcontroller was used to control the four wavelength illuminations provided by the PPG sensor and capture the reflected scattered light via a photodetector as shown in Figure 5. The multi-wavelength illumination was synchronized with sixteen LEDs, arranged as four channels of LEDs with four LEDs in each channel to provide the same wavelength illumination. The LEDs in each channel were driven in a cyclical sequence as the four LEDs in each channel were illuminated at the same time, when the LEDs in the other three channels were turned off. The signals captured from the individual wavelength illuminations of mOEPS were pre-amplified, then demultiplexed to determine each captured signal corresponding to the specified channel of wavelength illuminations, i.e., green, orange, red and IR. The captured signals were thus passed individually to be processed further as requested to extract relevant physiological parameters, i.e., SpO${}_{2}$.

The mOEPS system consists of (1) signal conditioning analogue front-end microelectronics, (2) digital signal processing unit (DSP), and (3) a communication port for either data storage or wirelessly transmitted to a PC for offline signal processing. All data sets obtained from the mOEPS were collected by the means of a 4-channel PPG board (DISCO4, Dialog Devices Ltd., Reading, Berkshire, UK). The analogue-to-digital conversion used to capture PPG signals was implemented on a 14-bit data acquisition board (DAQ, USB-6009, National Instruments Co., Novato, CA, USA) and the control software of the PPG board was implemented in a LabVIEW GUI (National Instruments Co., USA). A band pass filter range of 0.8 Hz to 5 Hz was implemented to reduce the influence of extraneous noise and a sampling frequency of 256 Hz was sufficient to reconstruct the PPG signals considering the Nyquist criteria.

## 3. Results

The comparative results of SpO${}_{2}$ (%) are presented to express the measurements using the commercial pulse oximetry, the green and orange illuminations and red and infrared illuminations of the mOEPS. The normality of the distributions of SpO${}_{2}$ (%) data was tested. A *t*-test was carried out on the SpO${}_{2}$ (%) data to find the correlations between the SpO${}_{2}$ (%) readings in each sub-protocol.

### 3.1. Oxygen Saturation: SpO${}_{2}$(%)

#### 3.1.1. First Sub-Protocol: Free Hand Movement

The green and orange illuminations from the mOEPS were used to extract SpO${}_{2}$, and their results were compared with those of a commercial pulse oximetry device (TempIRTM, Shenzhen Jumper Medical Equipment Co., Ltd., Shenzhen, China). Figure 6a (subject I) shows well-matched readings of SpO${}_{2}$ between mOEPS and pulse oximetry. However, for subject II in Figure 6b, there was a significant difference of three units with a single SpO${}_{2}$ reading of 98% from TempIRTM, and 95% from the mOEPS, respectively. The overall SpO${}_{2}$ readings, as shown in Figure 6c, for all 15 subjects show a high agreement with a correlation of *r* = 0.98.

The means from the Bland–Altman method [27] were used to evaluate the agreement of SpO${}_{2}$ readings between the mOEPS and the pulse oximetry. Firstly, the analysis was carried out with each subject to make sure that the data was matched and then the entire data of all 15 subjects were combined together to produce an overall agreement. Figure 7 demonstrates the Bland–Altman plots for all 15 subjects.

A *t*-test was carried out on the SpO${}_{2}$ data and a strong relationship was found between the SpO${}_{2}$ measured by using the mOEPS and the pulse oximetry, and vice versa (*r* = 0.98). Figure 8 shows the box-and-whisker plots for subject I, II and 15 subjects, respectively. As per Figure 8, a significant difference can be seen with the reading of subject II. However, when all the data sets of SpO${}_{2}$ for all subjects were aggregated, the overall box-and-whisker plot is comparable and shows no significant difference with value (*p* = 0.88), as shown in Figure 8.

The comparison study between the standard pulse oximetry (TempIRTM) and the green and orange illuminations of mOEPS was performed to obtain SpO${}_{2}$ readings. The results of SpO${}_{2}$ indicate that both measurements readings have a very similar range of variation ($\sigma^{2}$ = ±1). The mean difference for the both techniques is 0.03 as anticipated in a higher correlation. The star sign (*) identifies the readings of SpO${}_{2}$ that are of a very low value due to a subject’s sweaty skin during the experiment implementation.

#### 3.1.2. Second Sub-Protocol: Physical Movement

The results of this protocol were extracted by using mOEPS wavelength illuminations. The SpO${}_{2}$ reading obtained from green and orange illuminations were compared with red and infrared illuminations during physical exercises. Figure 9 shows the overall SpO${}_{2}$ readings for all 16 subjects in cycling and walking. The result shows well-matched readings of SpO${}_{2}$ between green/orange and red/infrared in both exercises with a correlation of (*r* = 0.98).

The agreements of SpO${}_{2}$ readings obtained from green/orange and red/infrared were validated by using the Bland–Altman plot. Figure 10a,b demonstrates the Bland–Altman plots for cycling and walking for all 16 subjects.

The comparison results of mOEPS was performed to obtain SpO${}_{2}$ readings from different illuminations. The results of SpO${}_{2}$ indicate that both measurements readings have a very similar range of variation ($\sigma^{2}$ = ±1) with the probability value consistently being 0.88. The star sign (*) identifies the readings of SpO${}_{2}$ that are of a very low value due to a subject’s sweaty skin during the experiment implementation.

## 4. Discussion

The mOEPS sensor provides multi-wavelength illumination sources to enable optoelectronic sensor based physiological monitoring more optional on signal processing to extract vital signs, i.e., SpO${}_{2}$. The experimental performance has demonstrated that the measurements with the mOEPS were less affected by small movements compared to those with the commercial pulse oximetry. For example, if the incident light from one LED, or even one channel of LEDs, becomes misaligned with the capillary arteries, at least one of the other source lights aligns with the same lay of the sensor; thus, the mOEPS is still likely to have sufficient alignment with the area of interest to be able to capture a useful reflected signal.

Referring to Figure 2, there are differences in the absorption of Hb and HbO${}_{2}$ at the illuminated wavelengths identified as green and orange, making these better suited for the extraction of SpO${}_{2}$ values. At these wavelengths, the penetration into the skin is significantly shallower compared with red and IR illuminations, implying that the use of green and orange illuminations for SpO${}_{2}$ can work during free hand movement. The subjects as implemented in the first sub-protocol, felt free to do anything, such as typing, pointing, holding and making unconscious hand movements during speaking, unlike some restrictive procedures and setups for measuring SpO${}_{2}$ from present pulse oximetry devices.

The *t*-tests were applied in this study to determine whether there was a significant difference between means of the mOEPS and SpO${}_{2}$ worked out. The correlation and the Bland–Altman plot were also utilized to evaluate the agreement between the measurement techniques in the both sub-protocols. The SpO${}_{2}$ readings from mOEPS were validated against the values of SpO${}_{2}$ obtained from the pulse oximeter in the first sub-protocol, whereas mOEPS was only used in the second sub-protocol to verify the SpO${}_{2}$ worked out from two set of illuminations. During the resting mode, the results of SpO${}_{2}$ readings were found to be in a higher agreement, as shown in Figure 6, with *r* = 0.98. The correlation of cycling and walking showed a good agreement of *r* = 0.98 as shown in Figure 9. As depicted in the Bland–Altman plot in Figure 7 and Figure 10, the comparison of the SpO${}_{2}$ from both designated sub-protocols are presented, where the points represent individual readings of SpO${}_{2}$.

All data readings of SpO${}_{2}$ are gathered together and displayed in the overall box-and-whisker plots. The results are comparable and there is no significant difference, with the value of *p* = 0.88, as shown in Figure 8. However, there was a three unit difference between pulse oximeter and mOEPS reading of SpO${}_{2}$ for subject II due to sweaty skin during the experiment.

The SpO${}_{2}$ readings are in the acceptable range of B ± 1.96 SD of the differences between the Pulse oximeter and the mOEPS signals as well as differences between both sets (green/orange and red/infrared) of mOEPS signals.

The average of the bias difference is close to zero and most of the points fell within the upper and lower limit while some points were presented out of the range due to different readings between the commercial device and mOEPS. These points are out of the line regressions and acceptable range, which indicates that there are some variations of these readings between the Pulse oximeter and the mOEPS due to fact that both sensors might have misaligned with area of interest. In addition, the SNR of Pulse oximeter is decreased with movements.

Table 2 and Table 3 represent the SpO${}_{2}$ readings that imply that both measurements of both sub-protocols have a very similar range of variation $\sigma^{2}$ = ±1. The outcome from the present study suggests that both green and orange illuminations of mOEPS are practicable for extracting SpO${}_{2}$ readings during free hand movement and physical activity.

## 5. Conclusions

The green and orange illuminations of mOEPS demonstrated that the readings range of SpO${}_{2}$ (98–99%) can be extracted in a range of physiological monitoring application scenarios. The results of SpO${}_{2}$ extracted from the mOEPS have demonstrated, (1) an excellent correlation *r* = 0.98 between the responses of green and orange illuminations, and the pulse oximetry with a statistical test *p* = 0.88; and (2) a same quality correlation *r* = 0.98 between the responses of green and orange illuminations, and red and IR illuminations in the same mOEPS with a statistical test *p* = 0.88.

Due to the nature of multi-wavelength illuminations in the mOEPS, illumination adaption can be performed in the mOEPS to obtain SpO${}_{2}$ readings with a certain degree of tolerance to different skin types and skin thicknesses [25]. The forthcoming study to validate SpO${}_{2}$ readings from the green and orange illuminations from mOEPS will be carried out in compliance with the ISO 80601-2-61, 2017 [28] and the FDA Pulse Oximeters—Premarket Notification Submissions [510(k)s], 2017 [29]. The study will be working closely with clinical professionals along with these standards in a gas controllable chamber/hypoxic condition against a clinical golden standard of blood gas analyzer for SpO${}_{2}$ reading accuracy.

Thus, the mOEPS could be consolidated with currently available wearable and smart devices for forthcoming real-time physiological monitoring to satisfy the requirements of present clinic monitoring and assessment. In addition, the mOEPS would appear to be of value in applications of sport physiological monitoring and assessment during physical movements, and is also likely to have its applications in personal healthcare, where it can provide essential vital signs with an indeed cost-effective solution.

Furthermore, one observation from the current study is that the mOEPS could provide an alternative opto-physiological monitoring solution that would be well suited to incorporation into wearable devices.

As part of the study of a wearable solution, the simplification of the controlling electronics and the embedding data processing algorithms will be considered. Several challenges remain to be overcome for the mOEPS through improvements in packaging, automated adaption to different subjects and the provision of users with recommendations for suitable operational procedures to satisfy requirements of cost-effective physiological monitoring in individuals or clinical settings.

## Figures and Tables

**Figure 1 sensors-19-00118-f001:**
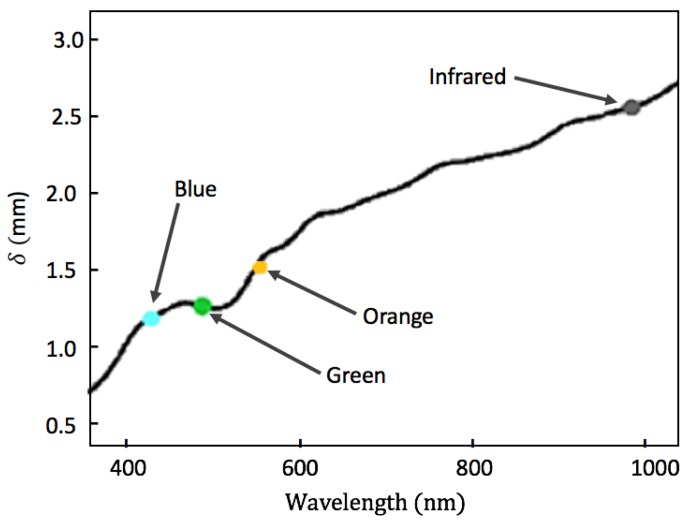
Optical penetration depth δ shown for a range of light sources of different wavelengths [21].

**Figure 2 sensors-19-00118-f002:**
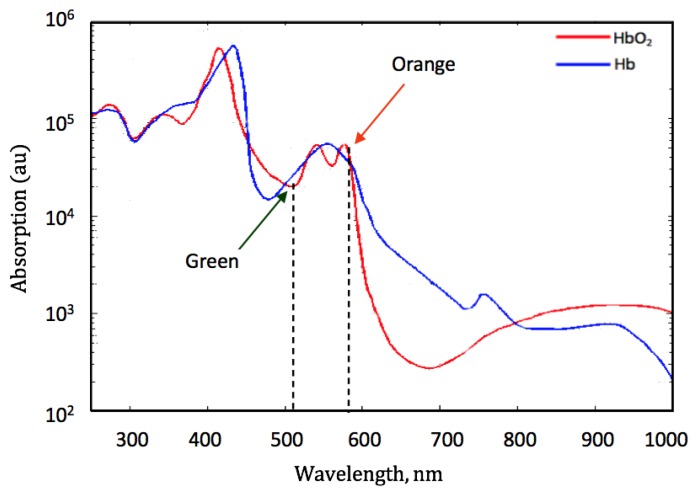
Absorption of Hb and HbO${}_{2}$ at green and orange illuminations [22].

**Figure 3 sensors-19-00118-f003:**
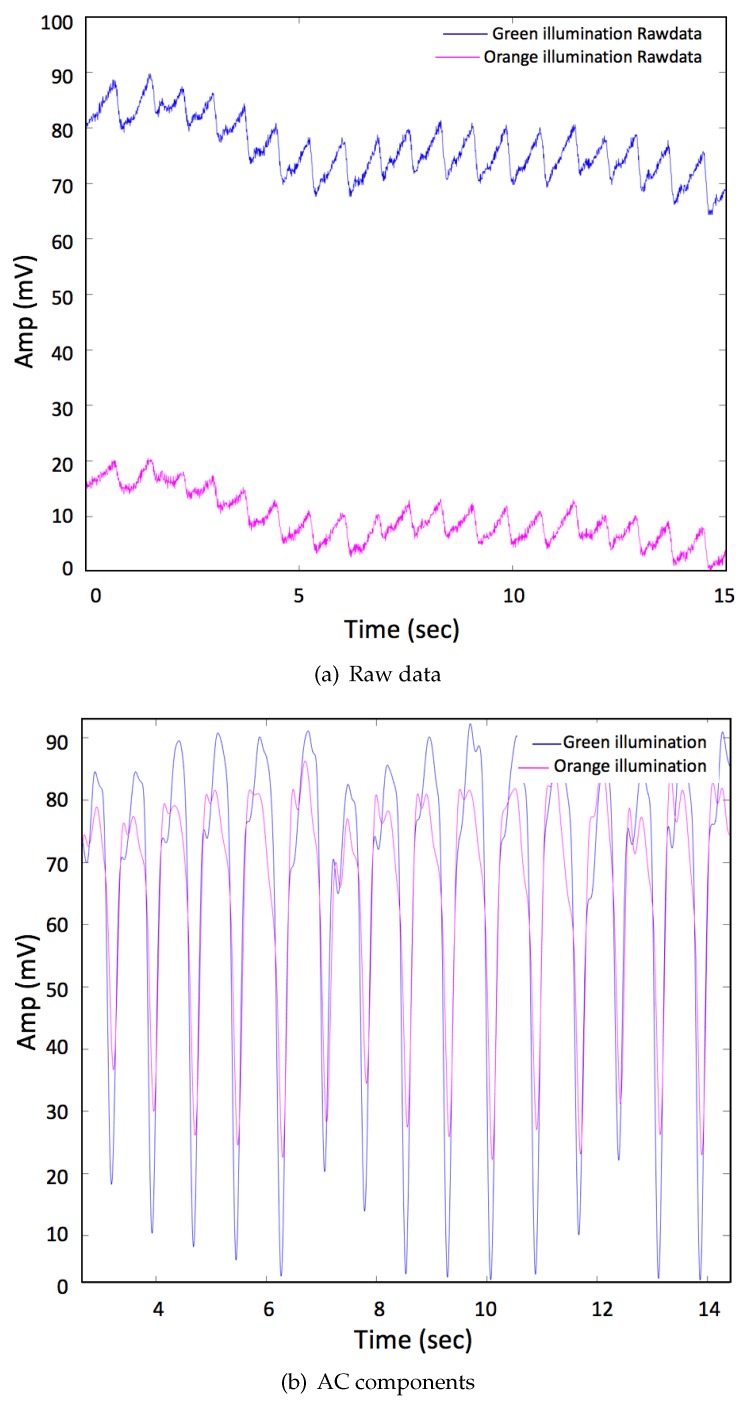
Data obtained from the mOEPS system (**a**) the raw green and orange signals; (**b**) the signals from the four wavelength illuminations after the band pass filtering (0.8–5 Hz). The peaks and troughs in the signals can be used to obtain SpO_2_.

**Figure 4 sensors-19-00118-f004:**
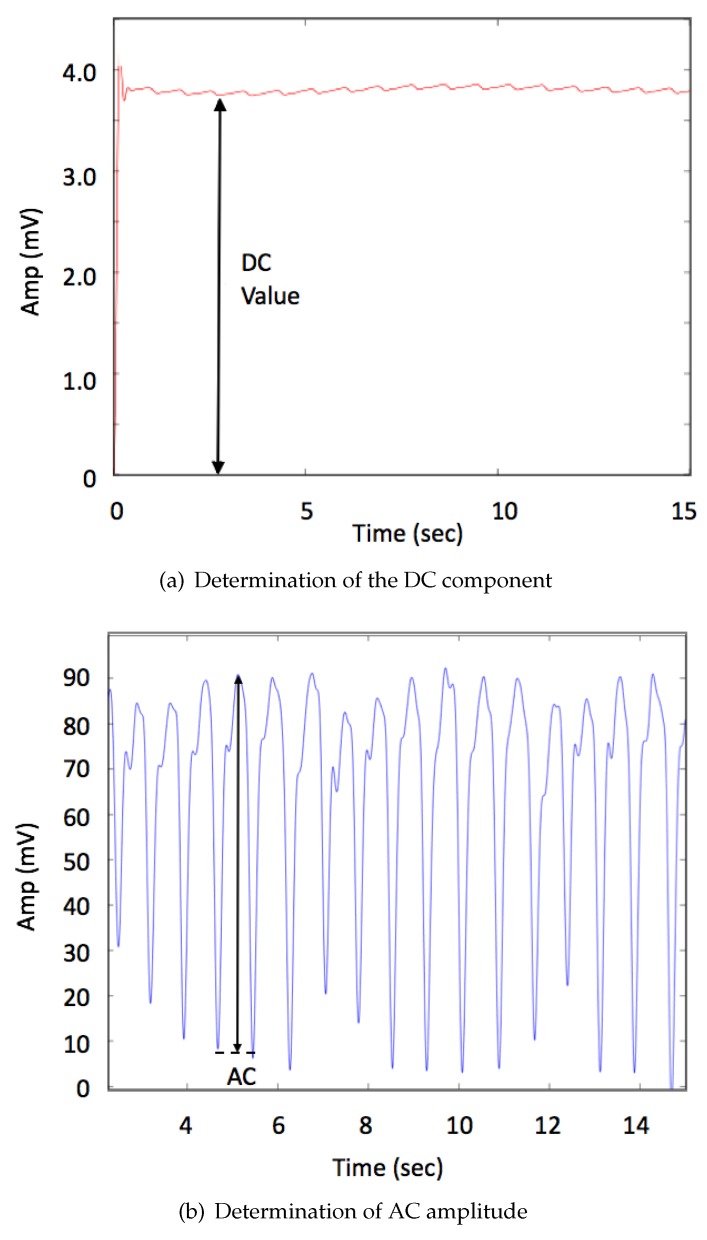
Measurement of PPG signals to extract the DC and AC components. (**a**) the raw data contains both information of non-pulsatile (DC) and a relatively small variation resulting from pulsatile (AC) absorption; (**b**) filtering of the raw data can be used obtain the dynamic AC change.

**Figure 5 sensors-19-00118-f005:**
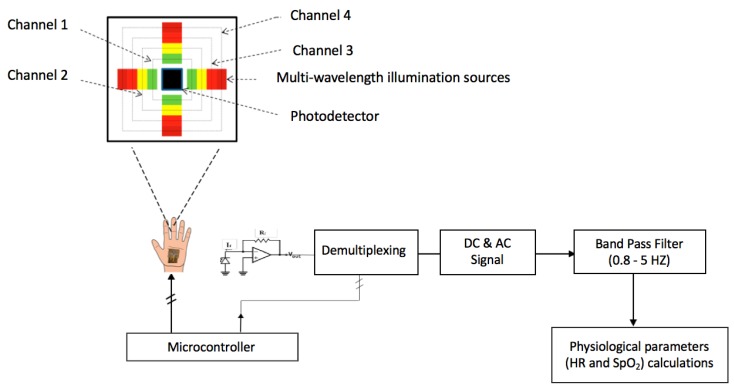
Microcontroller control of the multi-wavelength illumination and de-multiplexing operations. In the mOEPS, the detail of four illumination are: the channel 1: green, the channel 2: orange, the channel 3: red, and the channel 4: near infrared(IR).

**Figure 6 sensors-19-00118-f006:**
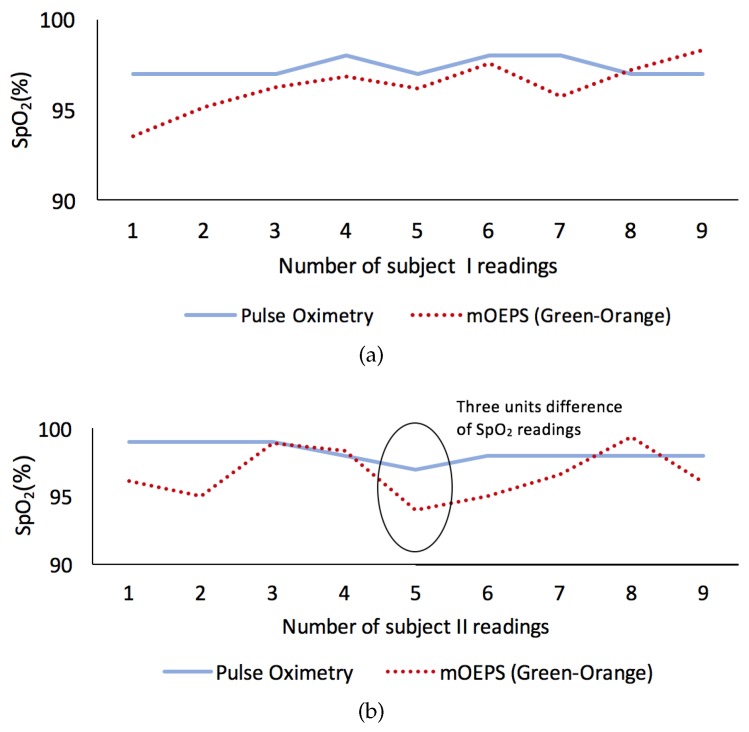
SpO_2_ data obtained from the mOEPS and a standard pulse oximetry device. The graphs of (**a**–**c**) display a line correlation for subject I, subject II and all 15 subjects respectively.

**Figure 7 sensors-19-00118-f007:**
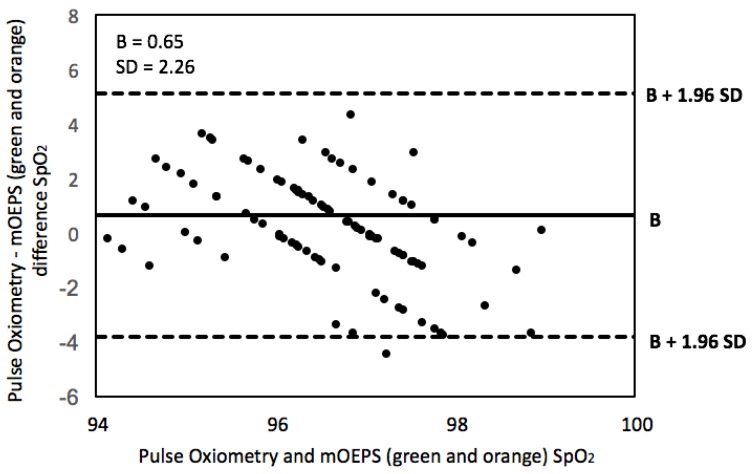
Bland–Altman plot showing differences in SpO${}_{2}$ between the green and orange illuminations of mOEPS and the pulse oximetry device. The graphs display a scatter diagram of the differences against the averages of the pulse oximetry and the green and orange illuminations of mOEPS for all 15 subjects.

**Figure 8 sensors-19-00118-f008:**
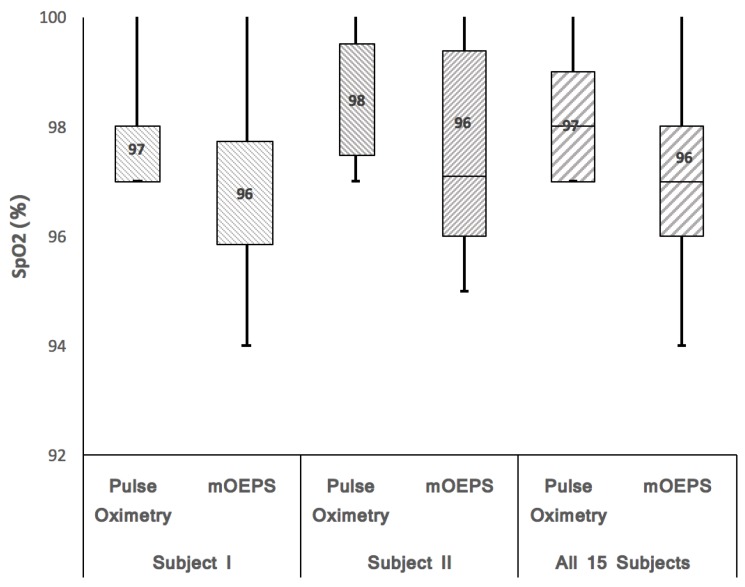
SpO${}_{2}$ readings with box-and-whisker plots and median values of the pulse oximetry device and the green and orange illuminations of mOEPS. Subject I (Pulse Oximetry: 97% and mOEPS: 96%), subject II (Pulse Oximetry: 98% and mOEPS: 96%) and all 15 subjects readings (Pulse Oximetry: 98% and mOEPS: 98%). All SpO${}_{2}$ data sets were processed with box-and-whisker procedures to obtain the lower, median and higher values.

**Figure 9 sensors-19-00118-f009:**
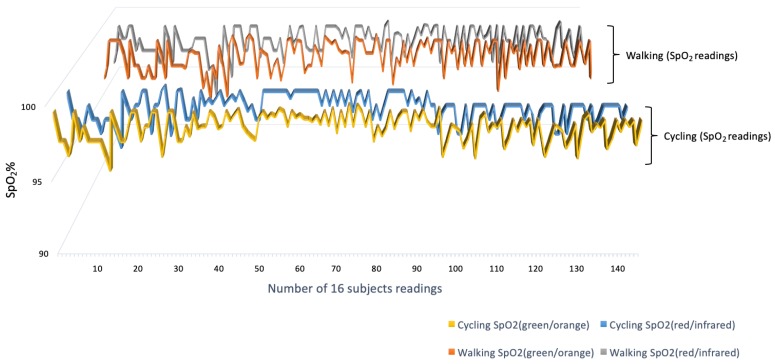
SpO${}_{2}$ data obtained from the mOEPS (green/orange) and the mOEPS (red/infrared) for different exercises.

**Figure 10 sensors-19-00118-f010:**
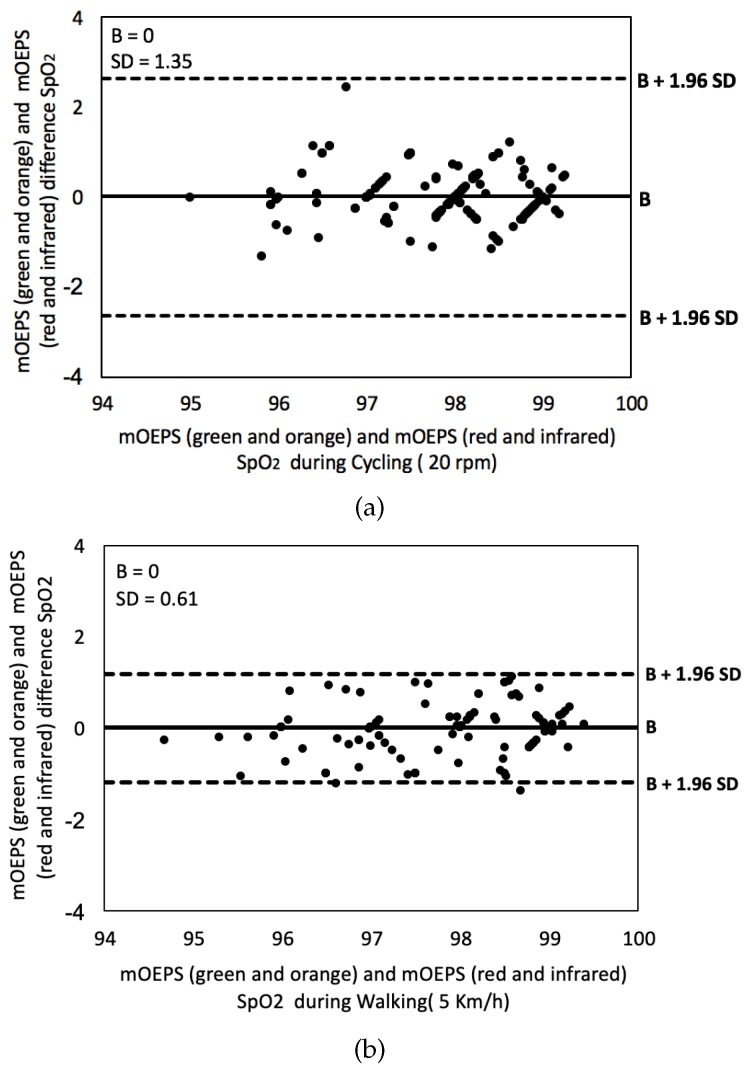
Bland–Altman plot showing differences in SpO${}_{2}$ between the green and orange illuminations of mOEPS and the red and infrared illuminations of mOEPS for (**a**) cycling and (**b**) walking. The graphs display a scatter diagram of the differences against the averages of the green/orange and red/infrared illuminations of mOEPS for 16 subjects.

**Table 1 sensors-19-00118-t001:** The values of Ratio (*R*) and SpO${}_{2}$(%) used in the calculation of the coefficients A and B.

Ratio of Green/Orange in mOEPS	SpO${}_{2}$ (Pulse Oximetry)
0.153	98
0.156	98
0.150	98
0.176	97
0.168	97
0.170	97
0.190	96
0.150	99
0.147	96
0.151	99
0.154	98
0.143	99
0.148	99
0.153	99

**Table 2 sensors-19-00118-t002:** Typical SpO${}_{2}$ (percentage points) comparison between the standard pulse oximetry and the green and orange illuminations of mOEPS. The star sign (*) identifies the readings of SpO${}_{2}$ that are of a very low value due to a subject’s sweaty skin during the experiment implementation.

Pulse Oximetry	mOEPS (Green/Orange)	Variation (Percentage Points)	Pulse Oximetry	mOEPS (Green/Orange)	Variation (Percentage Points)
94	95	−1	96	93 *	3
93 *	93 *	0	95	96	−1
96	97	1	96	96	0
97	97	0	97	98	−1
96	96	0	97	97	0
96	95	1	97	98	−1
96	96	0	97	98	−1
96	92 *	4	97	97	0
98	97	1	96	95	1
98	97	1	95	94	1
98	98	0	98	98	0
97	97	0	97	96	1
97	96	1	97	94	3
97	95	2	97	96	1
94	94	0	97	97	0
95	90	5	97	97	0
95	95	0	96	96	0
97	97	0	96	96	0
97	98	−1	97	96	1
97	97	0	98	97	1
96	97	−1	97	96	1
97	98	−1	98	98	0
97	93 *	4	97	97	0
97	98	−1	97	98	1
97	97	0	99	99	0

**Table 3 sensors-19-00118-t003:** Typical SpO${}_{2}$ (percentage points) comparison between the green/orange and red/infrared illuminations of mOEPS.The star sign (*) identifies the readings of SpO${}_{2}$ that are of a very low value due to a subject’s sweaty skin during the experiment implementation.

Cycling			Walking		
mOEPS (Green/Orange)	mOEPS (Red/Infrared)	Variation (Percentage Points)	mOEPS (Green/Orange)	mOEPS (Red/Infrared)	Variation (Percentage Points)
99	99	0	97	97	0
98	98	0	98	98	0
98	98	0	98	98	0
98	98	0	98	98	0
99	99	0	98	98	0
98	98	0	95	95	1
98	98	0	96	96	1
98	98	0	94	93 *	1
99	99	0	97	97	0
99	99	0	96	96	0
98	97	1	99	99	0
98	98	0	97	97	0
99	98	1	95	95	0
98	97	1	99	99	0
99	99	0	99	99	0
98	98	0	99	99	0
95	96	−1	97	97	0
98	98	0	97	97	0
99	99	0	99	99	0
98	98	0	99	99	0
98	98	0	99	99	0
97	98	−1	96	96	0
99	99	0	98	98	0
98	97	1	98	98	0
97	96	1	98	98	0
99	99	0	98	99	−1

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
