# Peer review of "Oxygen Saturation Measurements from Green and Orange Illuminations of Multi-Wavelength Optoelectronic Patch Sensors"

_sensors, 2018, doi:10.3390/s19010118_

Round 1

Reviewer 1 Report

The authors have compared the performance green and orange illumation based pulse oximeter to conventional red and infra-red ilumination based pulse oximeter.

I have couple of comments prior to publication:

1 ) Equations (2) and (3): Beer-Lambert's law shows an exponential decay of the light intensity when light passes through a medium. Commonly, the absorption coefficient µ  and path length d are considedered as a positive and thus there should be minus sign in the exponent (I =I_{in}*exp(-µd) ). This might have a effect also to Eqs. (4) and (5).

2) When defining a correlation, your null hypothesis can be H0: r=0 and alternative hypothesis r!=0 ("r not equal to 0") or r>0 or r<0. For this statistical testing, you get some statistical metrics, including critical values of the test variable that can be utilized to find out the p-values for the hypothesis testing. The authors have not reported any values (e.g. p-values) on the statistical significance on the correlation coefficients.

3) Tables 2-3: In metrology, the measurement error is a difference of the measurement result and the reference value, i.e. the error can be negative if the value of the reference is larger than the measurement result. I wonder why the authors report the absolute value of the error in Tables 2 -3 instead of error (which they call variation). If the error has a sign, a reader will quickly see if the results were too large or too small compared with the reference.

4) Figure 8: For subjects I and II, the range of mOEPS variesfrom 94% to 98% or 99%, but for all subjects, the mOEPS is only from 97% to 99%. Please explain why, i.e. where have you left the datapoints below 97% in all subjects' data which should contain also subjects I and II ??

5) The caption of Fig. 10b: "Km/h" should be "km/h".

6) The authors use quite many times the following notation: "variation σ= ±1". Why they use letter σ which may refer to standard deviation? In this case, it does not refer to a standard deviation but it refers to the difference of the measurement result and a reference value (i.e. measurement error)?

7. Lines 255-257:

"The results of SpO2 show that green and orange illuminations of mOEPS have the greatest modulation with the value of correlation r = 0.98 and p = 0.88 regardless of sitting or free hand movement. " Please rewrite the cited sentence, i.e. which statistical test p=0.88 referes. Currently, it is unclear where it refers (i.e. In the current form, p=0.88 can be interpreted as a p-value of correlation 0.98 which it most likely is not.)

8) There are 2 types of statistical tests: parametric and non-parametric. Parametric tests are intended for normally distributed data and non-parametric for data having any distributions. The authors implement parametric t-tests but they do not discuss the normality of their data at all, not in the manuscript or in the previous reviewer response.

Author Response

Answered the questions as attached

Reviewer 2 Report

This paper proposes the use of green/orange wavelengths in recording reflectance photoplethysmography (PPG) signals to estimate blood oxygen saturation level (SpO2). The motivation to use green/orange wavelengths, as noted in the paper, is a less sensitivity to motion artefacts affecting PPG signals for example during physical exercise. Although this is a good motivation, the results do not completely support the claims. Based on this, I have two major concerns.

1- The exercise has been performed under normal conditions in this research. In an old study:

- H. Benoit, "Accuracy of pulse oximetry during intense exercise under severe hypoxic conditions", 1997.

hypoxic conditions are proposed in order to make various levels of SpO2 (e.g. 70%-100%). This condition is not explored in this paper since all SpO2 levels are well above 90%. This can affect the interpretation of accuracy.  

2- The benchmark data has been collected using TempIRTM pulse oximetry. Pulse oximeters usually have an error of 2-3% for estimation of SpO2 at rest condition. During the exercise, pulse oximeters might not be accurate enough to estimate SpO2. The authors can provide detailed information from the device whether they have used regarding human controlled desaturation studies and been conducted at rest/exercise.   

Minor comments:

- The last plot of Figure 6 is a good plot with promising results (probably not the two first plots). The axis labels are a bit confusing e.g. Number of 15 subjects readings.

- In Figure 7, the number of points considering one subject is limited, therefore; not the best presentation using bland-altman. 

- Figure 9 is also a good figure. However, this figure shows that red/infrared also produce good estimates for SpO2. The authors can compare green/orange wavelengths to red/infrared PPG signals in great details and see how the signal quality index can be different for different wavelength under physical exercise and rest.

- The authors might have a look at: http://www.nihonkohden.de/uploads/media/SpO2-Report_03.pdf

for SpO2 monitoring: a pulse oximeter accuracy study.

- There are missing papers in the references. The authors should refer to recent papers for SpO2 estimation using reflectance PPGs.

Author Response

answered the questions as attached.

Reviewer 3 Report

The paper is well structured and written. The results are well supported by the method employed. Maybe figures with the data could be made as vertical panels rather than horizontal ones and bigger so the reader can much appreciate the example traces

Author Response

Appreciated the comments as attached.

Round 2

Reviewer 2 Report

The authors have answered my major comments; have not applied. Overall, the paper is interesting since it tests green and orange wavelength, however, the validation has been performed versus a platform which can be inaccurate. Although as the authors have replied, the errors are acceptable in clinical setting [but mainly during rest] to infer about clinical outcomes in an inference based system; from a validation point of view and considering physical exercise, the results cannot be used to make a profound conclusion regarding the system accuracy.  

Still I should note that the work is interesting and can make a big impact for healthcare applications but in the current format there is not enough evidence. Optical sensors are very sensitive to motion, considering the benchmark data, TempIRTM pulse oximetry (Shenzhen Jumper Medical Equipment Co. Ltd, China), I am not sure how accurate is the benchmark under physical exercise to be used as a reference for validation. Also, inclusion of hypoxic conditions can help.

Author Response

Please see the answers in the attachment.

Round 3

Reviewer 2 Report

I accept the paper because if its potentials for future work.